# It takes more than a machine: A pilot feasibility study of point-of-care HIV-1 viral load testing at a lower-level health center in rural western Uganda

**Ross M. Boyce**[1,2]*, **Ronnie Ndizeye**[3], **Herbert Ngelese**[3], **Emmanuel Baguma**[3], **Bwambale Shem**[4], **Rebecca J. Rubinstein**[2], **Emmanuel Rockwell**[5], **Sarah C. Lotspeich**[5], **Bonnie E. Shook-Sa**[5], **Moses Ntaro**[3], **Dan Nyehangane**[6,7], **David A. Wohl**[1], **Mark J. Siedner**[8], **Edgar M. Mulogo**[3]

1 Institute for Global Health and Infectious Diseases, University of North Carolina at Chapel Hill, Chapel Hill, North Carolina, United States of America, 2 Department of Epidemiology, Gillings School of Global Public Health, University of North Carolina at Chapel Hill, Chapel Hill, North Carolina, United States of America, 3 Faculty of Medicine, Department of Community Health, Mbarara University of Science & Technology, Mbarara, Uganda, 4 Bugoye Level III Health Center, Uganda Ministry of Health, Kasese District, Uganda, 5 Department of Biostatistics, Gillings School of Global Public Health, University of North Carolina at Chapel Hill, Chapel Hill, North Carolina, United States of America, 6 Epicentre Mbarara Research Centre, Mbarara, Uganda, 7 Faculty of Medicine, Department of Medical Laboratory Science, Mbarara University of Science & Technology, Mbarara, Uganda, 8 Massachusetts General Hospital and Harvard Medical School, Boston, Massachusetts, United States of America

* roboyce@med.unc.edu

**Data Availability Statement:** Deidentified individual data that supports the results will be shared following publication provided the

## Abstract

Barriers continue to limit access to viral load (VL) monitoring across sub-Saharan Africa adversely impacting control of the HIV epidemic. The objective of this study was to determine whether the systems and processes required to realize the potential of rapid molecular technology are available at a prototypical lower-level (i.e., level III) health center in rural Uganda. In this open-label pilot study, participants underwent parallel VL testing at both the central laboratory (i.e., standard of care) and on-site using the GeneXpert HIV-1 assay. The primary outcome was the number of VL tests completed each clinic day. Secondary outcomes included the number of days from sample collection to receipt of result at clinic and the number of days from sample collection to patient receipt of the result. From August 2020 to July 2021, we enrolled a total of 242 participants. The median number of daily tests performed on the Xpert platform was 4, (IQR = 2–7). Time from sample collection to result was 51 days (IQR = 45–62) for samples sent to the central laboratory and 0 days (IQR = 0–0.25) for the Xpert assay conducted at the health center. However, few participants elected to receive results by one of the expedited options, which contributed to similar time-to-patient between testing approaches (89 versus 84 days, p = 0.07). Implementation of a rapid, near point-of-care VL assay at a lower-level health center in rural Uganda appears feasible, but interventions to promote rapid clinical response and influence patient preferences about result receipt require further study.

**Trial registration:** ClinicalTrials.gov Identifier: NCT04517825, Registered 18 August 2020. Available at: https://clinicaltrials.gov/ct2/show/NCT04517825.

investigator who proposes to use the data has approval from an Institutional Review Board (IRB), Independent Ethics Committee (IEC), or Research Ethics Board (REB), as applicable, and executes a data use/sharing agreement with UNC. Researchers may apply for data access by contacting the corresponding author or the UNC IRB at irb_questions@unc.edu.

**Funding:** The study was funded by a Development Award from the University of North Carolina at Chapel Hill Center for AIDS Research, an NIH funded program (Grant P30 AI050410) to RMB, who also acknowledges support from a Caregivers at Carolina Award made by the Doris Duke Charitable Foundation (Award 2015213). Xpert HIV VL cartridges were supplied by Cepheid at no cost to the study. Database support was provided by the North Carolina Translational and Clinical Sciences (NC TraCS) Institute, which is funded by the National Center for Advancing Translational Sciences (NCATS), through Grant Award Number UL1TR002489. The funders had no role in study design, data collection and analysis, decision to publish, or preparation of the manuscript.

**Competing interests:** Xpert HIV VL cartridges were supplied by Cepheid at no cost to the study. There are no patents, products in development or marketed products associated with this research to declare. This does not alter our adherence to PLOS ONE policies on sharing data and materials.

## Background

Viral load (VL) monitoring is associated with earlier detection of treatment failure, more timely and appropriate initiation of second-line antiretroviral therapies (ART), and better outcomes for people living with HIV (PLHIV) [1–3]. For these reasons, VL monitoring is an essential component of HIV care. Scale-up of VL testing has been a global priority following release of the World Health Organization's (WHO) consolidated ART guidelines in 2013, which first recommended using VL measurement instead of CD4 counts [4]. While substantial gains have been made across sub-Saharan Africa (SSA), logistical and technical barriers continue to limit progress towards key global targets [5,6].

In 2015, Uganda adopted VL testing as the preferred approach for monitoring response to ART [7]. Despite this policy change, the vast majority of PLHIV in Uganda live in rural areas and receive care at lower-level health facilities where VL testing is not available on site. Instead, VL testing is accomplished via a network of "hubs," which collect samples from nearly 3,000 neighboring facilities and forward them on to the Ugandan National Health Laboratory Services (UNHLS) in Kampala. While a centralized approach to VL monitoring consolidates laboratory infrastructure and technical expertise, there remain a number of disadvantages and potential pitfalls to this strategy, including (i) loss or spoilage of specimens in transit and (ii) disruptions in the transportation networks that may contribute to delays between testing and receipt of results.

Furthermore, when VL results are ≥1,000 copies/ml, current Ugandan treatment guidelines recommend monthly intensive adherence counseling (IAC) for three months followed by another VL test, which if still unsuppressed should prompt a regimen switch [7]. Given the stepwise process, delays in the return of results may substantially increase the risk of inaction and thereby limit the potential benefits of VL monitoring. These factors may partly explain the lower rates of viral load suppression (VLS) seen in many rural areas of SSA [8–10].

Over the past decade, there has been a revolution in molecular technology that has resulted in robust diagnostic tools that can be performed outside of traditional laboratories, often described as near point-of-care (NPoC) [11]. One of the pioneering devices is the GeneXpert platform with the MTB/RIF assay (Cepheid Inc. Sunnyvale CA, USA), which detects *M. tuberculosis* as well as the mutations that confer resistance to first-line therapies [12,13]. These platforms, distributed to a network of nearly 150 laboratories, are now utilized for routine tuberculosis (TB) across Uganda [14]. However, multiple studies have shown that device implementation alone did not reduce TB-associated mortality, the proportion of people who had initiated treatment, nor the time to treatment for people with a positive test result, even though detection rates were increased [15,16]. These unexpected findings were largely attributed to weaknesses in the existing health system such as high rates of loss to follow-up and delayed time to treatment initiation.

More recently, an assay for quantification of HIV-1 viral load (Xpert VL) using the same GeneXpert platform has been introduced. The accuracy of the Xpert VL assay has been shown to be comparable to reference assays in frontline settings [17]. Moreover, reports from closely-monitored studies in larger health facilities and urban settings have demonstrated both the feasibility of implementation and promise of improved clinical outcomes with NPoC VL testing [18,19]. Therefore, the overarching objective of this study was to determine whether the ancillary systems and processes required to realize the potential of rapid molecular technology are available at a prototypical lower-level facility, (i.e., primary health center) in rural western Uganda–and if not, identify the key gaps and barriers to inform future trials. Our hypothesis was that implementation of NPoC HIV-1 testing without accompanying modifications to clinic flow, laboratory processes, and clinical protocols, would not result in significant improvement in clinical outcomes in PLHIV.

## Material and methods

### Study site

The study was conducted in Bugoye sub-county located in the highlands of western Uganda ([Fig 1]). The prevalence of HIV in the Mid-West region, where Bugoye is located, is estimated to be 5.7% (95% confidence interval [CI] 4.7–6.8) among adults 15 to 64 years of age, which is broadly similar to the national estimate (6.2%, 95% CI 5.8–6.7) [20]. However, rates of VLS in the region are modestly lower than the national rate as well as that observed in more urban areas such as Kampala. The sub-county's primary public health facility is the Bugoye Level III Health Center (BHC), which is comprised of a 25-bed inpatient ward, a busy outpatient clinic, a maternity ward, and a small laboratory. One day per week, there is a differentiated ART clinic for PLHIV with a census of approximately 500 active clients. The clinic is supervised by a clinical officer, with a level of training approximating that of a physician assistant in the United States, and a nurse. The ART clinic is the only such public facility in the sub-county with walking times from some of the areas along the boundary of the Rwenzori Mountains National Park ranging up to three hours. However, as a Level III Health Center, BHC does not have on-site HIV VL testing. Instead, dried blood spots (DBS) are collected and sent by motorcycle to the nearest hub site, approximately 25 km away, from where they are forwarded on to UNHLS.

### Study design

The study was a two-phase, open-label pilot designed to assess routine process measures after implementation of a rapid molecular HIV-1 VL testing infrastructure at a level III health

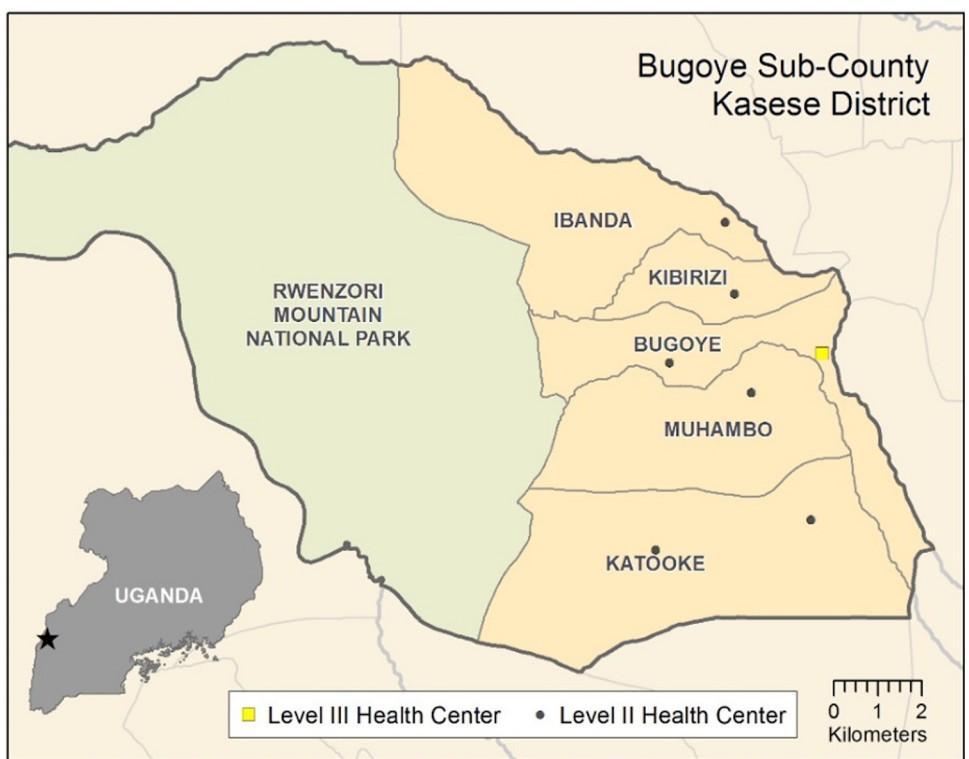

**Fig 1. Map of study area in reference to Uganda.** Shapefile obtained from GADM.org (https://geodata.ucdavis.edu/gadm/gadm4.1/shp/gadm41_UGA_shp.zip). License information amd terms of use available at https://gadm.org/license.html.

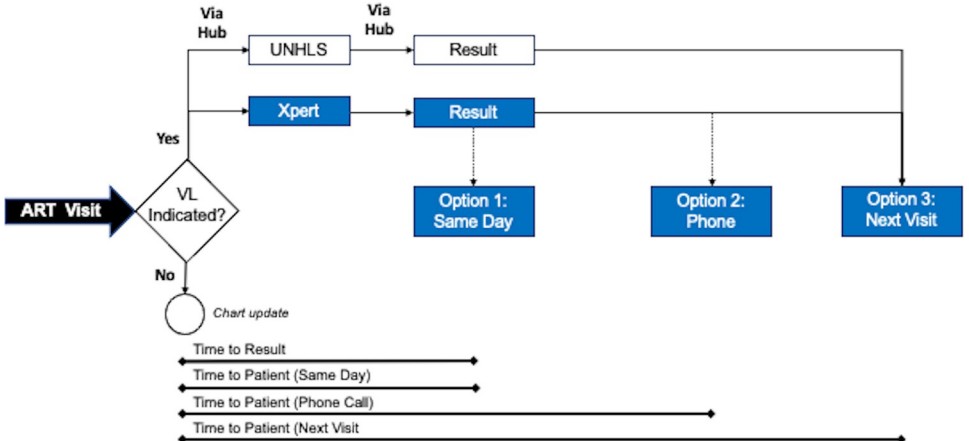

**Fig 2. Flowchart illustrating parallel viral load testing and study outcomes in Phase 2.**

center in rural western Uganda. The first three months of the study were focused on participant enrollment, and collection of baseline data from the clinic. During Phase 1, VL testing was performed in accordance with the existing standard of care (i.e., samples sent to the UNHLS for testing), while the GeneXpert platform was installed and staff training was completed. In the subsequent nine months, herein Phase 2, VL testing was performed in parallel to routine care using the on-site GeneXpert platform and associated HIV-1 VL assay. The primary outcome was the number of VL tests successfully performed each clinic day. Secondary outcomes included additional domains of feasibility that would be required to realize the potential of NPoC VL testing [21] including (i) the number of days between sample collection and the result returning to the clinic (i.e., the time-to-result), and (ii), number of days between sample collection and the patient receipt of the result (i.e., time-to-patient). We also assessed measures of acceptability including the proportion of patients electing to receive results by waiting at the clinic or phone call (**Fig 2**). Further evaluation of acceptability was elicited through participant interviews and focus-group discussions, which will be reported separately. All adult patients (age ≥18 years) receiving care for HIV at the BHC ART clinic were eligible to participate. Clinical decision making remained the responsibility of the attendant Clinical Officer, who received study results, but were directed to rely on the UNHLS results for decision making.

## Recruitment and enrollment

Each clinic day, study staff fluent in the local dialect provided a summary of study objectives, methods, and eligibility criteria to patients in the waiting room of the ART clinic, after which time attendees were provided the opportunity to ask questions in the group setting. Individuals expressing interest were then moved to a private area where staff reviewed the study documents and answered additional questions. Those individuals agreeing to participate provided written consent. Recruitment continued on a rolling basis through the study period.

## Study encounters

After consent was provided, study staff administered a questionnaire detailing the participant's demographic and health history (**S1 File**). Additional information was abstracted directly from the clinic registers including duration of clinic enrollment, results of previous laboratory

testing, prior ART regimens, and history of opportunistic infections, if available. Participants progressed through the various clinic stations (e.g., waiting room, clinician visit, laboratory, and pharmacy) in a manner similar to that of non-participants. If the attendant Clinical Officer determined that VL testing was indicated, either at the initial or any subsequent visit, laboratory staff collected the required sample according to the study phase. In Phase 1, laboratory staff collected a dried blood spot (DBS), which was sent to the UNHLS, while in Phase 2, staff collected both a DBS and approximately 5 ml of venous blood, which was subsequently centrifuged and the resulting plasma tested on site using the Xpert HIV-1 VL assay. Results of VL testing were entered into the study database immediately after testing was complete or when results returned from the UNHLS. Participants undergoing VL testing were offered a choice between receiving results by phone or at the next scheduled clinic visit. In Phase 2, when NPoC testing was available, participants were also able to elect to wait at the clinic to receive results (i.e., "same day") and undertake any further intervention if not suppressed.

## Laboratory testing

Since May 2016, routine measurement of HIV VL has been performed at UNHLS in Kampala with samples shipped via a network of "hub" facilities available in all 112 districts of Uganda [22]. In brief, after each clinic day blood samples from patients requiring VL testing are collected on filter paper, packaged together, and transported by motorcycle to the district hospital in Kilembe, approximately 25 kilometers from Bugoye. There, DBS from multiple peripheral clinics are collected and shipped by bus to the UNHLS for testing. During the study period, testing at UNHLS was performed using the Abbott Real Time HIV-1 PCR Quantitative assay. Paper-based print-outs are then returned to the clinic utilizing the same network. Participants receive their results at the next visit, which is typically scheduled in 3 to 6 months for established (i.e., ≥12 months in care) patients with a suppressed VL, but monthly for patients who are not suppressed [7].

The Xpert HIV-1 VL assay performed on the GeneXpert instrument system, is an *in vitro* diagnostic test designed for the rapid quantification of HIV-1 in human plasma from individuals with an active HIV infection. It uses real-time quantitative reverse transcription PCR technology with a limit of quantitation of 40 copies/ml the ability to detect HIV-1 RNA over a linear range of 40 to 10,000,000 copies/ml. The GeneXpert platform allows on-demand molecular testing in one fully integrated closed cartridge and provides results in 90 minutes [23]. Multiple studies have shown a high degree of correlation between results obtained from the Xpert HIV VL assay and various reference assays [24]. All cartridges were obtained directly from the manufacturer and used prior to the expiration date. Each day, staff recorded the total number of tests performed, the number of tests failures and/or invalid tests, and equipment downtime related problems with software or power supply.

Installation of the Gene Xpert IV took place during Phase 1. The unit was placed in a small, secure room in a previously unused building that had been recently renovated as part of an ongoing research partnership at the site (**Fig 3**). On the advice of colleagues from the local district hospital, which maintains a GeneXpert IV for tuberculosis diagnosis, we installed an air conditioning unit to prevent extreme air temperature and dust that might negatively impact operation of the unit. We also purchased and installed a centrifuge, which is not routinely available at most level III health centers, to obtain plasma from collected whole blood. In September 2020, a team led by the manufacturer's in-country representative along with a co-investigator (DN) from Epicentre Mbarara Research Centre conducted a two-day, practical training program for study staff and laboratory technicians from the health center.

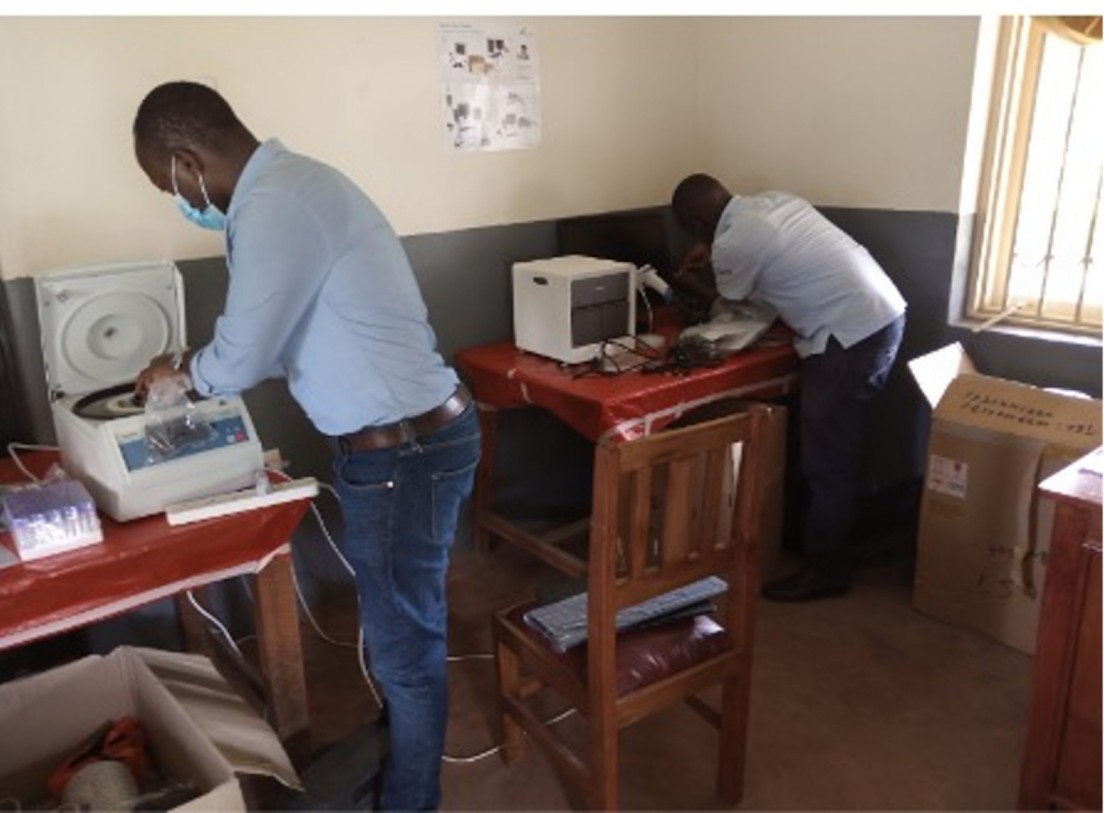

**Fig 3. Installation of the GeneXpert platform at BHC.**

## Statistical analysis

Study data were recorded in REDCap [25]. a secure electronic database using tablet devices equipped with internet connectivity. All statistical analyses were performed using R version 4.1.1 [26]. Descriptive statistics were stratified by test type and computed for participant demographics, years on ART, and ART regimen. To measure time-to-result and time-to-patient, we calculated the number of days from the date of the clinic visit until lab results were received by the clinic and delivered to the patient, respectively. Median time-to-result and time-to-patient were estimated for the two test approaches, with associated confidence intervals based on unadjusted ordinary least-squares (OLS) modelling. Additionally, median time-to-result and time-to-patient were calculated stratified by patients' feedback preferences to control for the effect of different modes of feedback delivery. We also assessed routine performance of the GeneXpert assay by quantifying differences in VL results compared to the routine UNHLS. Furthermore, we categorized the VL results into three categories that align with relevant clinical thresholds of (i) not detectable, (ii) less than 1,000 copies/ml, (iii) and greater than 1,000 copies/ml in order to assess potential impacts on clinical decision-making between the two test types. Due to the pilot nature of the study, no power calculations were conducted for the primary or secondary outcomes [27,28].

## Ethical approvals

Study procedures were approved by the University of North Carolina Institutional Review Board (19–2363), the Mbarara University of Science and Technology Research Ethics Committee (16/07-19), and the Uganda National Council of Science and Technology (HS 2756).

## Results

From August 2020 to July 2021, we enrolled a total of 242 participants (**Table 1**). Participants had a median age of 37 years (IQR 30–45), and nearly three quarters were women (n = 175, 72.3%). Most participants (n = 204, 97%) were established in clinic and already taking ART at the time of enrollment. The most common ART regimens were TDF/3TC/DTG (n = 135, 55.8%) and TDF/3TC/EFV (n = 61, 25.2%). Most participants (145 of 211, 68.7%) had experienced at least one ART regimen switch with changes from TDF/3TC/EFV (n = 82, 56.6%) or AZT/3TC/NVP (n = 52, 35.9%) being most frequent. Among patients with at least one VL result from UNHLS documented in the medical record at enrollment, 83 of 94 (88.3%) had achieved VLS, defined as a result of <1,000 copies/ml.

During the study period, attendant providers ordered a total of 111 tests, including 24 tests in Phase 1 when only routine testing was available, and 87 tests in Phase 2, when the Xpert assay was available on-site but samples continued to be sent to the UNHLS in parallel. In Phase 1, which overlapped with strict COVID-19 lockdowns [29], a total of 8 tests were

**Table 1. Demographic and clinical characteristics of the study cohort.**

| Study Variables | Total |
|---|---|
| | (N = 242) |
| **Age (Years)** | |
| Mean (SD) | 37.9 (10.5) |
| Median [Min, Max] | 37.0 [18.0, 76.0] |
| Missing | 6 (2.5%) |
| **Gender** | |
| Female | 175 (72.3%) |
| Male | 62 (25.6%) |
| Missing | 5 (2.1%) |
| **Marital Status** | |
| Divorced | 7 (2.9%) |
| Married | 160 (66.1%) |
| Not married | 53 (21.9%) |
| Separated | 6 (2.5%) |
| Widowed | 11 (4.5%) |
| Domestic partnership | 1 (0.4%) |
| Missing | 4 (1.7%) |
| **Time on ART (Years)** | |
| Mean (SD) | 5.07 (3.61) |
| Median [Min, Max] | 5.00 [0, 15.0] |
| Missing | 33 (13.6%) |
| **ART Regimen** | |
| ABC/3TC/ATV | 2 (0.8%) |
| AZT/3TC/EFV | 1 (0.4%) |
| AZT/3TC/NVP | 5 (2.1%) |
| TDF/3TC/ATV | 2 (0.8%) |
| TDF/3TC/DTG | 135 (55.8%) |
| TDF/3TC/EFV | 61 (25.2%) |
| TDF/3TC/LOP/r | 3 (1.2%) |
| AZT/3TC/ATV | 1 (0.4%) |
| Missing | 32 (13.2%) |

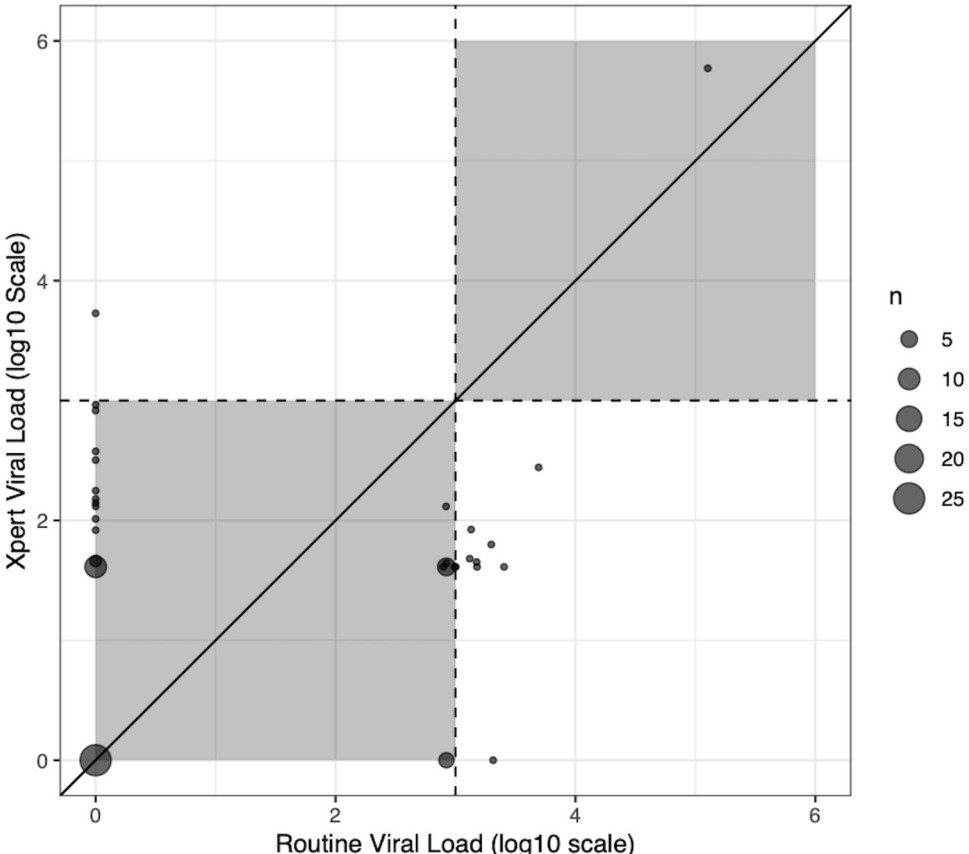

Note: points near the diagonal line nearly agree in exact VL measurement and points in the shaded regions outlined by the dashed lines agree in clinical VL measurement.

**Fig 4. Results of paired Xpert and routine HIV VL testing.** Clinically relevant threshold of 1,000 copies/mL indicated by dashed lines. Points falling within the same quadrant would represent cases where clinical management would be same based on either Xpert or routine result.

completed at UNHLS and returned to BHC (2.6/month). In Phase 2 a total of 80 tests were completed at UNHLS and returned to BHC, while 87 tests were successfully performed on the GeneXpert platform at BHC (18.6/month). A total of 33 ordered tests were not completed because participants did not stay to have blood drawn, including 4 in Phase 1 and 29 in Phase 2. The number of daily tests performed on the Xpert platform ranged from 1 to 18 (median = 4, IQR 2–7).

Among paired results, agreement between routine and rapid testing was high when considering the relevant threshold of ≥1,000 copies/ml, at which point further clinical intervention (e.g., intensive adherence counseling, regimen switch) would be indicated (**Fig 4**). The most frequent source of discordant test results were values of <1,000 copies/ml on the Xpert platform, but ≥1,000 copies/ml on routine testing. The absolute magnitude of the difference between these discordant results, however, was relatively modest (mean Log = 0.113, 95% CI: -0.262–0.488). A total of 8 of 96 (8.3%) VL tests that were performed on the Xpert platform were recorded invalid results. Most invalid results (n = 5) were attributable to power interruptions mid-assay. Two invalid results were due to insufficient sample being added to cartridge and one was caused by a poorly processed sample (i.e., mixed with red cells).

For UNHLS testing, the median time-to-result was 51 days (IQR 45–62) between sample collection and the return of the result to the health center across both study phases. Notably,

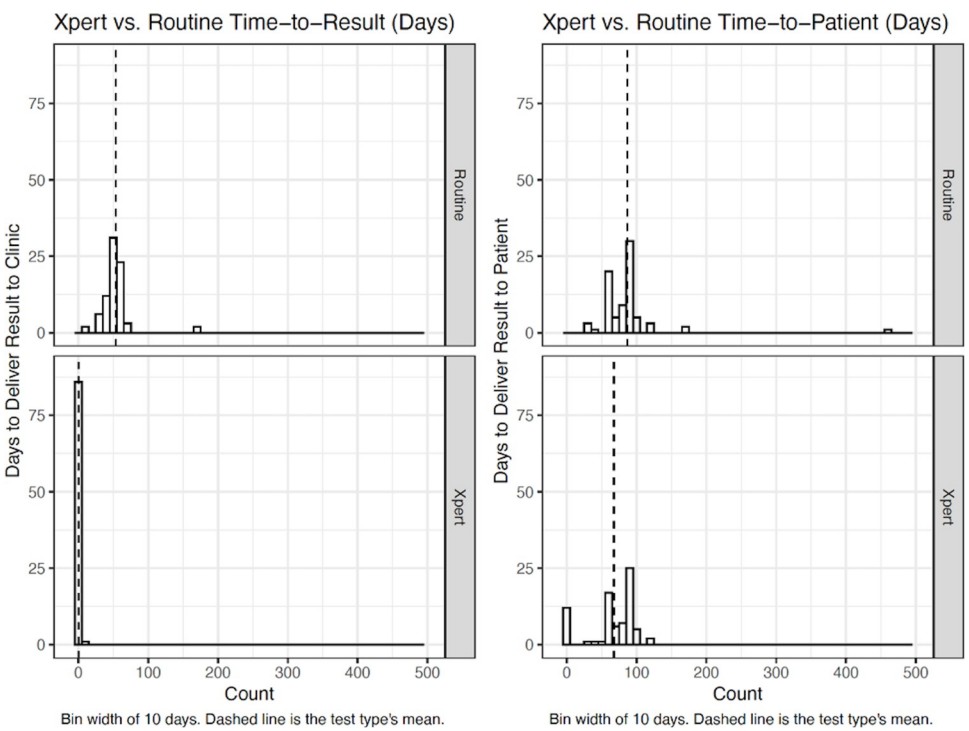

**Fig 5. Histograms showing time-to-result and time-to-patient for each testing option.**

seven samples were submitted to the UNHLS but had not returned by the conclusion of the study, five of which had been outstanding for ≥120 days. In contrast, more than three-quarters of valid results from the Xpert assay (n = 68, 78.2%) were available the same day (median time-to-result = 0 days, IQR 0–0.25, p<0.001). In Phase 2, when participants were asked how they would like to receive their results, the overwhelming majority of participants (66 of 77, 87.4%) elected to receive results at the next visit. The median time-to-patient was therefore similar between UNHLS and the Xpert approaches (89 versus 84 days, p = 0.07), even though in Phase 2 most results were available within 24 hours of sample collection (Fig 5). Only eight partici-pants chose to remain at the clinic for their results and three elected to receive results by phone. All those who elected to wait received their results the same day, while those who requested a phone call received results within 24 hours.

## Discussion

Our study–to our knowledge the first to be conducted at a level III health center—demon-strates that implementation of a rapid molecular platform to perform HIV-1 VL testing at peripheral health facilities in rural western Uganda is feasible. Despite the limited infrastruc-ture available, we did not observe utility disruptions or supply shortfalls that substantially impacted operation of the GeneXpert platform. Most notably, our results show significant reductions in the time-to-result compared to the current standard of care. However, similar to reports of Gene Xpert implementation for TB testing, we did not find that the HIV-1 viral load Xpert assay reduced time to patient receipt of results. This suggests that additional interven-tions will be needed to enable prompt patient engagement and provider action for this assay to improve clinical care. Overall, these findings provide important preliminary evidence that

further decentralization of VL testing may be one component of efforts to improve rates of VLS in rural areas of SSA.

The marked difference in time-to-result between the Xpert assay and routine UNHLS testing is not surprising given the inherent logistical demands of the centralized hub network. Yet the magnitude of the difference, approximately 50 days, was unexpected. This finding is similar to results from the RAPID-VL study that took place in 20 PEPFAR-funded clinics across Southwestern Uganda, where the median turnaround time was reduced by 55 days following GeneXpert implementation [30]. These results clearly demonstrate the potential of rapid molecular platforms to impact the delivery of HIV care primarily through increased patient engagement, higher uptake of VL testing, and more timely clinical decision making.

In contrast, our results also demonstrate that without changes in existing clinic processes and patient preferences, the advantages of an on-site rapid molecular platform may be negated. As evidence of this, we observed that the vast majority of patients elected not to receive results via one of the expedited options, which led to a similar time-to-patient between testing methods. Ideally, VL results could be returned to patients on same date, as was achieved in the STREAM trial [31]. A number of participants in our study, however, expressed an aversion to remaining at the clinic to receive their VL results. While not stated overtly, this sentiment may reflect longstanding stigma at being seen at the ART clinic. It may also represent low knowledge among participants—as well as the providers—regarding the importance of VL testing. One relatively simply intervention may be rearranging clinic workflows, which currently relegate phlebotomy and laboratory testing as the final step before discharge. With prior review of scheduled attendees, patients due for VL testing could have blood drawn on arrival, thereby reducing potential wait times. Other mechanisms to facilitate timely and effective communication may also be needed. The RAPID-VL intervention, for example, paired rapid molecular testing with same/next-day telephone delivery of results [30]. Similar approaches, informed by local stakeholders, are needed to determine the optimal method of return of results in rural areas, where mobile phone ownership remains lower than urban areas, particularly among women [32].

The observed rate of invalid Xpert HIV VL tests, approximately 8%, is notable, particularly given concerns regarding the cost of the cartridge compared to current methods. The majority of invalid results (5 of 8, 62.5%) were due to power interruptions during the sample runs. This issue could be addressed with back-up power systems, recognizing this would likely require additional investment. A smaller proportion of invalid results could be attributed to poor sample preparation. We are reassured that the invalid rate due to non-electrical issues is similar to that observed in other studies from SSA, many of which were conducted in more advanced laboratory settings [18,33,34]. However, in any discussion of the invalid rate, we would highlight that a similar number of samples sent to UNHLS (n = 7, 7.4%) never had results return to the health center and were assumed to be either lost or spoiled in transport. This is an important consideration, as clinic staff often wait months before determining that another sample needs to be sent to UNHLS, whereas as invalid result on the Xpert platform is immediately recognized and could be repeated without causing substantial delays in clinical decision making.

Our study has a number of strengths the most remarkable of which was implementation at a lower-level, rural health center where there has been very little research conducted in the ART clinic to date. This setting provided valuable experience and insight into unanticipated barriers that may be representative of the challenges faced with more widespread implementation. At the same time, the single site and relatively small number of VL tests limits the generalizability of our results. Additionally, the duration of the pilot study did not allow us to follow participants long enough to assess for potential impacts on rates of VLS or other improvements in health resulting from implementation. The relatively short duration and pilot nature

of the study also limited our ability to assess supply chain issues, as cartridges were obtained in bulk and directly from the manufacturer. Lastly, much of the study took place early in the global COVID-19 pandemic, including periods of highly enforced travel restrictions in Uganda, which may have reduced care seeking and clinic volumes, thereby limiting the demand on lab staff and equipment performing VL testing.

## Conclusions

Implementation of a rapid, near point-of-care VL assay at a lower-level health center in rural Uganda appears feasible, but interventions to promote rapid clinical response and influence patient preferences about result receipt require further study. Given further evidence from higher-level facilities suggesting that implementation could be a key strategy to improving rates of VLS and achieving the UNAIDS targets, additional efforts are needed to expedite decentralized testing.

## Supporting information

**S1 File. Data collection forms for baseline and follow-up visits.**
(PDF)

## Acknowledgments

We wish to thank the individuals who participated in the study for their contributions and feedback without which this study would not have been possible. In addition, we acknowledge the generous and always gracious support of the Bugoye Leve IIl Health Centre and staff.

## Author Contributions

**Conceptualization:** Ross M. Boyce, Bonnie E. Shook-Sa, Moses Ntaro, David A. Wohl, Mark J. Siedner, Edgar M. Mulogo.

**Formal analysis:** Ross M. Boyce, Rebecca J. Rubinstein, Emmanuel Rockwell, Sarah C. Lotspeich, Bonnie E. Shook-Sa.

**Funding acquisition:** Ross M. Boyce, David A. Wohl.

**Investigation:** Ross M. Boyce, Ronnie Ndizeye, Herbert Ngelese, Emmanuel Baguma, Bwambale Shem, Moses Ntaro, Dan Nyehangane, Edgar M. Mulogo.

**Methodology:** Ross M. Boyce, Bonnie E. Shook-Sa, Dan Nyehangane, David A. Wohl, Mark J. Siedner, Edgar M. Mulogo.

**Project administration:** Ronnie Ndizeye, Emmanuel Baguma, Moses Ntaro, Edgar M. Mulogo.

**Supervision:** Ross M. Boyce, Ronnie Ndizeye, Emmanuel Baguma, Bwambale Shem, Moses Ntaro, Dan Nyehangane, Edgar M. Mulogo.

**Visualization:** Emmanuel Rockwell, Sarah C. Lotspeich.

**Writing – original draft:** Ross M. Boyce, Emmanuel Rockwell.

**Writing – review & editing:** Ross M. Boyce, Ronnie Ndizeye, Herbert Ngelese, Emmanuel Baguma, Bwambale Shem, Rebecca J. Rubinstein, Emmanuel Rockwell, Sarah C. Lotspeich, Bonnie E. Shook-Sa, Moses Ntaro, Dan Nyehangane, David A. Wohl, Mark J. Siedner, Edgar M. Mulogo.

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
