## [Decision Letter · Decision Letter 0]

21 Nov 2022

PGPH-D-22-01433

It takes more than a machine: A pilot feasibility study of health systems capacity building for point-of-care HIV-1 viral load testing in rural western Uganda

Dear Dr. Boyce,

Thank you for submitting your manuscript to PLOS Global Public Health. After careful consideration, we feel that it has merit but does not fully meet PLOS Global Public Health’s publication criteria as it currently stands. Therefore, we invite you to submit a revised version of the manuscript that addresses the points raised during the review process.

Thank you for your submission which the reviewers and I read with interest. We feel it fits our journal however both reviewers have raised some material points which need to be addressed; most importantly about the aligning of the phrasing of study objectives, study design and title. 

We look forward to receiving your revised manuscript.

Kind regards,

Sabine Hermans

Academic Editor

Journal Requirements:

2. Our staff editors have determined that your manuscript is likely within the scope of our Diagnostics in Global Health Call for Papers. This editorial initiative is headed by a team of Guest Editors for PLOS GPH: Senjuti Saha (Child Health Research Foundation, Bangladesh) and Titus Divala (Public Health Scotland, University of Glasgow and University of Malawi College of Medicine). The Collection will encompass a diverse range of research articles about diagnostics in global health, including innovation and deployment of point of care diagnostics; subsets of diagnostics related to infectious diseases, chronic diseases and injuries; policies related to and regulation of diagnostics; supply chain issues; and the affordability, accessibility, and availability of essential diagnostics.  Additional information can be found on our announcement page: https://collections.plos.org/call-for-papers/diagnostics-in-global-health/

3. Please send a completed 'Competing Interests' statement, including any COIs declared by your co-authors. If you have no competing interests to declare, please state "The authors have declared that no competing interests exist". Otherwise please declare all competing interests beginning with the statement "I have read the journal's policy and the authors of this manuscript have the following competing interests:"

4. Please amend your detailed Financial Disclosure statement. This is published with the article. It must therefore be completed in full sentences and contain the exact wording you wish to be published.

a. State what role the funders took in the study. If the funders had no role in your study, please state: “The funders had no role in study design, data collection and analysis, decision to publish, or preparation of the manuscript.”

b. If any authors received a salary from any of your funders, please state which authors and which funders.

5. Please provide separate figure files in .tif or .eps format only and remove any figures embedded in your manuscript file. Please also ensure that all files are under our size limit of 10MB.

6. Figure 1: please (a) provide a direct link to the base layer of the map (i.e., the country or region border shape) and ensure this is also included in the figure legend; and (b) provide a link to the terms of use / license information for the base layer image or shapefile. We cannot publish proprietary or copyrighted maps (e.g. Google Maps, Mapquest) and the terms of use for your map base layer must be compatible with our CC-BY 4.0 license. 

7. Figure 3 includes an image of an identifiable person. Please provide written confirmation or release forms, signed by the subject(s) (or their parent/legally authorized guardian), giving permission to be photographed and to have their images published under our CC-BY 4.0 license. 

Otherwise, we kindly request that you remove the photograph.

8. We have noticed that you have uploaded Supporting Information files, but you have not included a list of legends. Please add a full list of legends for your Supporting Information files after the references list.

Additional Editor Comments (if provided):

Reviewers' comments:

Reviewer's Responses to Questions

**Comments to the Author**

1. Does this manuscript meet PLOS Global Public Health’s publication criteria? Is the manuscript technically sound, and do the data support the conclusions? The manuscript must describe methodologically and ethically rigorous research with conclusions that are appropriately drawn based on the data presented.

Reviewer #1: Yes

Reviewer #2: Yes

2. Has the statistical analysis been performed appropriately and rigorously?

Reviewer #1: Yes

Reviewer #2: Yes

3. Have the authors made all data underlying the findings in their manuscript fully available (please refer to the Data Availability Statement at the start of the manuscript PDF file)?

Reviewer #1: Yes

Reviewer #2: No

4. Is the manuscript presented in an intelligible fashion and written in standard English?

Reviewer #1: Yes

Reviewer #2: Yes

5. Review Comments to the Author

Reviewer #1: 1. The title requires minor edits

2. The background of the study in not clearly described as to why it is relevant - what was happening the BHC to warrant this pilot study? That is missing out

3. Phase should have incorporated pre-COVID-19 UNHLS data so that the pandemic impact would be well elucidated

4. I have attached comments on the PDF document which can help your team of authors to polish the document

5. Update the discussion session with the highlights flagged out in the results section

6. Make the conclusion concise based on the results and discussion - looking the title can help firm up your conclusion

Reviewer #2: This is a rather interesting study on the opportunity for using Xpert for POC VL monitoring. It contributes to the current body of knowledge on the feasibility and acceptability of POC monitoring. Though, there are some major points where I feel improvement is warranted.

-The rationale of the study is not clear to me. I believe there is quite some literature showing feasibility already

-This project is called a pilot-feasibility study, though I think that is the wrong terminology. Pilot-feasibility studies are studies done to establish whether a clinical trial can be done. In such a case, you look at a number of factors such as being able to obtain enough participants? Number of times an intervention was used? Are all procedures for the trial in place? I would rather call this kind of an 'assessment to establish feasibility of POC monitoring in a rural cinic....' unless it was done in preparation of a trial, which would be mostly interesting, but then that should be described as the rationale

-I feel more information is needed on the validity with the used sample size. The size is small and it is questionable to related results

-The specific objectives or secondary objectives in the introduction are nog clear. The overarching objective is described would (I believe) to a qualitative design including observation and interviews with health care workers. Though a totally different methodology is used which I believe does not answer the overarching objective. It does answer the primary and secondary objectives though, so I believe the overarching objective should be adapted.

Methods:

-The objectives mentioned on page 6 are very smart and understandable. But I do not see how they will answer to the overarching objective stated at the end of the introduction on page 4.

-How were the measures of feasibility chosen? If it is about feasibility of conducting a study, that should be stated more explicitly. I believe the mentioned measures are not so much about feasibility but an outcome of the performance of the test and procedures

-Line 149: How was the measure of acceptability chosen? To me 'electing results by waiting' is not directly a measure of acceptability. There are many theoretical models for both acceptability and feasibility that could have been used. If such theories are not used, it should be clearly discussed why not and why the chosen measures were used

-More details are needed about the amount of blood collected

Results

-Table 1: Why do you display both the mean and the median?

Discussion

-Line 299: I do not see how this conclusion can be drawn from the results. The results do not say anything about modest infrastructure improvement and staff training. This was not measured

-More discussion is needed on possible implementation strategies to decrease time-to-patient for POC

-Discussion is needed on how the study can contribute to the current WHO guidelines which state that enhanced adherence counseling should be done at 50 copies (previously 1000 copies). In addition, I have been informed that most East African countries currently move to a different treshold varying from 50-200 copies, so this should be discussed as the study is of high relevance for that.

Please, adapt title and abstract based on the above points as well

6. PLOS authors have the option to publish the peer review history of their article (what does this mean?). If published, this will include your full peer review and any attached files.

**Do you want your identity to be public for this peer review?** For information about this choice, including consent withdrawal, please see our Privacy Policy.

Reviewer #1: **Yes: **Dr. Steve Wandiga

Reviewer #2: No

---

## [Editor Report · Decision Letter 1]

10 Feb 2023

It takes more than a machine: A pilot feasibility study of point-of-care HIV-1 viral load testing at a lower-level health center in rural western Uganda

PGPH-D-22-01433R1

Dear Dr. Boyce,

We are pleased to inform you that your manuscript 'It takes more than a machine: A pilot feasibility study of point-of-care HIV-1 viral load testing at a lower-level health center in rural western Uganda' has been provisionally accepted for publication in PLOS Global Public Health.

Best regards,

Sabine Hermans

Academic Editor